# Nutrient Composition of Ovary, Hepatopancreas and Muscle Tissues in Relation to Ovarian Development Stage of Female Swimming Crab, *Portunus trituberculatus*

**DOI:** 10.3390/ani13203220

**Published:** 2023-10-15

**Authors:** Wenping Feng, Zeping Zhao, Jiteng Wang, Tao Han

**Affiliations:** Fishery College, Zhejiang Ocean University, 1 Haida South Road Changzhi Island Lincheng New Area, Zhoushan 316022, China; wenpingfeng@outlook.com (W.F.); zzp199808@126.com (Z.Z.); wangjiteng1971@gmail.com (J.W.)

**Keywords:** ovarian development, lipid and fatty acid, protein and amino acid, *Portunus trituberculatus*

## Abstract

**Simple Summary:**

To clarify the specific nutritional needs for ovary maturation of the commercially important crab *Portunus trituberculatus*, we compared the biochemical composition of the ovary, hepatopancreas, and muscle at five ovarian developmental stages, and also analyzed their relation to the ovarian developmental stage of *P. trituberculatus*. We found that the branched-chain amino acids, long-chain polyunsaturated fatty acids, and monounsaturated fatty acids accumulated in the ovary and hepatopancreas, and positively correlated with the ovary maturation stage. Also, we found an increasing tendency of carotenoid and phosphatidylcholine in phospholipid from the pre-developmental stage to the proliferative stage in the ovary, but not in the hepatopancreas and muscle, indicating their crucial role for oocyte development. Our study suggests the involvement of these compositions in the broodstock diet, which will promote the ovarian maturation of adult *P. trituberculatus* and ensure high-quality larval production.

**Abstract:**

The swimming crab *Portunus trituberculatus* is one of the most important economic species in China and its mature ovary often determines its commercial value and production. Although the ovary maturation of crustaceans is generally affected by exogenous nutrition, the specific nutritional needs of ovary maturation of *P. trituberculatus* are poorly understood. To this end, we collected the *P. trituberculatus* samples with five ovarian maturation stages and measured their biochemical composition of the ovary, hepatopancreas, and muscle at each ovarian developmental stage. We further analyzed their relation to the ovarian developmental stage of *P. trituberculatus* by principal components analysis (PCA). We found the levels of branched-chain amino acids, long-chain polyunsaturated fatty acids (LC-PUFA), and monounsaturated fatty acids (MUFAs) in the ovary and hepatopancreas increased during the ovary maturation process, and also passively correlated with ovarian developmental stage, which highlights the necessity of these specific nutrients for oogenesis and for improving the nutrient quality of crabs. In addition, we found an increasing tendency of carotenoid content and phosphatidylcholine in phospholipid in the ovary from the pre-developmental stage to the proliferative stage, but not in the hepatopancreas and muscle, which highlights the possible involvement of carotenoids during the rapid oocyte development process. Our study may provide valuable information for developing a suitable broodstock diet that promotes the ovarian maturation of adult *P. trituberculatus* and ensures high-quality larval production.

## 1. Introduction

The swimming crab *Portunus trituberculatus* is widely distributed from China to Japan and Korea [1]. Since the good flavor of its mature ovary, *P. trituberculatus* has become one of the most economically important species in China [2]. However, the fishing of *P. trituberculatus* has caused a significant decline in the wild population in recent years [3]. To protect the wild population and meet market needs, investigating the artificial breeding of *P. trituberculatus* is necessary. In general, the ovarian maturation process is a critical period for successful aquaculture, including for the crustacea [4,5]. Since the function of the mature ovary is to produce and periodically release oocytes, the quality of the mature ovary as well as its oocytes directly determines the development of a fertilized egg, which in turn affects the production of crustacea [4,5]. Since the nutrient requirements of crustacea change along with the ovarian maturation process [6], understanding the key determinants of ovarian maturation is a prerequisite for improving the reproduction and larvae survival of *P. trituberculatus*.

It is well-known that the gonadal development process of crustacea is influenced by critical nutrients. The nutrition intake of broodstocks determines the nutritive value of the ovary, hepatopancreas, and muscle of *P. trituberculatus*, which affects commercial quality [7] and the ovarian maturation process [8,9]. Some key nutrients, such as amino acids and fatty acids, have been proven to play important roles in regulating the reproductive performance of animals [10,11,12,13]. Among them, amino acids affect the taste and participate directly in the synthesis of reproductive protein and immune response [14]. Fatty acids, especially long-chain polyunsaturated fatty acids (LC-PUFA) and monounsaturated fatty acids (MUFAs) are beneficial to crab quality and determine the gonad maturation process, fecundities, and larval hatching rate of crustacea [15,16]. In addition, carotenoid affects the color of tissues, which is the key indicator of commercial quality in crabs [17,18]. It also accumulates in the oocytes as part of a major egg yolk protein, which is the source of catabolic energy substrates during ovary maturation and early embryo development [19,20].

The nutritional values of crustaceans vary among their gonads, hepatopancreas, and muscle tissues, and represent a dynamic process with gonadal development [21]. When an oocyte rapidly develops, the nutrients allocated to the muscle tissue will be reduced. Meanwhile, the reserved nutrients in the hepatopancreas are primarily utilized for metabolic activities in the ovary, such as vitelline synthesis, oocyte development, and egg formation [22]. Insufficient nutrient supply may cause adverse consequences for ovarian development [22]. Thus, in this study, to understand the specific nutritional needs of ovary maturation in *P. trituberculatus*, we examined the biochemical components such as approximate composition, lipid fraction, fatty acid, amino acid, and carotenoid content of the ovary, hepatopancreas, and muscle during the ovarian maturation of *P. trituberculatus.* Our findings will promote the development of artificial feed for broodstocks, to achieve better reproductive performance, commercial traits, and higher quality eggs in *P. trituberculatus.*

## 2. Materials and Methods

### 2.1. Ethics Statement

*Portunus trituberculatus* were collected in the East China Sea in China. They are not privately owned or protected in any way. Field studies did not include endangered or protected species. All experimental procedures on animals were in compliance with the guidelines of Zhoushan Government.

### 2.2. Sampling

To obtain the whole ovarian maturation stages of the crab, *P. trituberculatus* were randomly caught in the East China Sea (122°28′45″, 29°53′45″) from July 2021 to May 2022. In each maturation stage, at least 10 females with healthy-looking features and robust appendages [21] were randomly selected and carefully transported to the laboratory at Zhejiang Ocean University for further analysis.

### 2.3. Growth Traits, Gonadal Index, and Hepatopancreas Index

All individuals were assessed for four growth traits: carapace length (CL), carapace width (CW), body height (BH), and wet body weight (BW). These traits were measured by standard Vernier calipers (accuracy 0.01 mm) and digital weighing (accuracy 0.1 g), respectively.

Each crab was dissected, and the wet weight of the ovary and hepatopancreas were measured using a microbalance (accuracy 0.001 g). The weights of the ovary and hepatopancreas were used to calculate the gonadal somatic index (GSI, %) and hepatopancreas index (HSI, %) according to the following formulas, respectively.
GSI = (gonadal wet weight/body wet weight) × 100(1)
HSI = (hepatopancreas wet weight/body wet weight) × 100(2)

The samples were anesthetized on ice until the dissection. Then the ovary, hepatopancreas, and muscle were extracted and stored at −80 °C for further analyses.

### 2.4. Gonadal Development

The gonads of all crabs were treated with 4% formalin for fixation and preservation. The established techniques for paraffin histology, as described by Waiho et al. in 2017 [23], were employed. Cross-sections of the gonads (5 μm) were obtained and stained with Mayer’s hematoxylin and eosin. The gonadal development was categorized into five stages based on the observed variations in the relative abundance and size of germinal cells within the gonads.

### 2.5. Proximate Composition and Fatty Acid Analysis

The composition of the crabs was analyzed using the methods of AOAC (1995) [24]. Moisture content was determined by subjecting the crabs to lyophilization (LL1500, Thermo Scientific, Waltham, MA, USA) at a temperature of −110 °C. Crude protein and ash content were measured using an Auto Kjeldahl System (K355/K437, Buchi, Flawil, Switzerland) and a muffle furnace (KSW, Kewei, Beijing, China) at 550 °C, respectively. The crude lipid content was determined using the chloroform/methanol (2/1, *v*/*v*) extraction method following the procedure described by Folch et al. in 1957 [25]. The fatty acid profiles were subsequently analyzed using gas chromatography (GC7890B, Agilent Technologies, Santa Clara, CA, USA) according to the methodology outlined by Liu et al., (2021) [26].

### 2.6. Amino Acids Analysis

The samples of the ovary, hepatopancreas, and muscle were hydrolyzed, wherein acid hydrolysis and alkali hydrolysis were performed for 17 amino acids and tryptophan, respectively. Then, the amino acid was analyzed using an automatic amino acid analyzer (Hitachi L-8900, Chiyoda, Japan) according to the description of Shi et al., 2016 [27] and Khas et al., 2022 [28].

### 2.7. Triglycerides and Total Cholesterol Analysis

The tissue sample was pulverized using a ceramic mortar and mixed with pre-cooled saline in a ratio of 1:9 (w:v) to create a homogenate. The homogenate was then subjected to centrifugation at 2500 rpm for 10 min at 4 °C using a frozen centrifuge (Fresco17, Thermo Scientific, Waltham, MA, USA) to separate the supernatant for subsequent analysis. The concentrations of triglycerides (TG) and total cholesterol (T-CHOL) were determined using kits (Nanjing Jiancheng Bioengineering Research Institute, Nanjing, China), following the manufacturer’s instructions.

### 2.8. Phospholipid and Carotenoid Analysis

The phospholipid of the ovary, hepatopancreas, and muscle was analyzed using an HPLC system (1200 Series, Agilent, Santa Clara, CA, USA), according to the description of Hang et al., 2015 [29]. Cardiolipin, phosphatidylethanolamine, phosphatidylcholine, phosphatidylcholine, and phosphatidylserine (Sigma-Aldrich, St. Louis, MO, USA) were used as standards.

The analysis of carotenoids in each of the three tissues was conducted using a Waters liquid chromatography system (Waters 1525) equipped with a model 2996 photodiode array detector (PAD). The detailed analysis method used is referred to in the report of Bing et al., 2015 [30].

### 2.9. Data Analysis

All the data underwent a two-step transformation algorithm (Templeton, 2011) [31] to stabilize the variance and were assessed for assumptions of normality (Shapiro–Wilk test) and homogeneity of variance (Levene’s test). To figure out the statistical differences between each growth trait and the nutrient compositions of tissues among different ovarian developmental stages, the data of growth traits, proximate compositions, triglycerides, and total cholesterol contents, which followed a normal distribution among ovarian maturation stages, were analyzed using a one-way ANOVA, followed by Tukey’s multiple comparison test. To figure out the differences among the ovarian developmental stages and among the tissues, the data of amino acids, fatty acids, phospholipids, and carotenoids content that generally followed a non-normal distribution were analyzed using the Kruskal–Wallis test, followed by the Steel–Dwass multiple comparison test. In addition, Spearman’s rank correlation test was conducted between every two growth traits to determine their strength of association in each ovarian maturation stage. Relationships between ovarian developmental stages and the amino acid and fatty acid contents in the ovary, hepatopancreas, and muscle were examined by principal component analysis (PCA). Statistical analyses were conducted using SPSS 21 and R version 3.3.3 (R Core Team, 2016) [32].

## 3. Results

### 3.1. Ovarian Maturation Stages

Five distinct developmental stages of ovarian maturation were identified according to the color and size of the ovary (Figure 1). The initial stage (Stage I) was characterized by a translucent and smooth ovary and corresponded to immature females that only possessed Stage I oocytes (Figure 1A,F). In a Stage II ovary, creamy white ovarian tissues were visible, and both Stage I and Stage II oocytes were present (Figure 1B,G). Stage III ovaries were characterized by an increase in ovarian tissue size, which exhibited light yellow coloration, with predominantly Stage III oocytes (Figure 1C,H). During Stage IV of ovarian development, the ovary exhibited expansion, and orange lobes were observed, covering approximately 50% of the dorsal hepatopancreas (Figure 1D,I). At Stage V, the ovary displayed a vibrant orange coloration and occupied more than half of the dorsal space within the carapace. As female crabs progressed to ovarian maturation in Stages IV and V, the size of oocytes increased. Enlarged hexagonal Stage IV oocytes became the predominant cells, characterized by a visible nucleus and yolk granules (Figure 1E,J).

### 3.2. Growth Traits, GSI, and HSI

The growth traits, gonad somatic index (GSI), and hepatopancreas somatic index (HSI) were compared among ovarian maturation stages (Table 1). Body weight (BW) and HSI increased before the ovary premature stage, then decreased during maturation (*p* < 0.05). However, GSI significantly increased throughout the maturation process. GSI and HSI were positively correlated with the ovary developmental stage (*p* < 0.05) (Figure 2).

### 3.3. Proximate Composition

The protein values in the hepatopancreas significantly decreased, while the protein and lipid values in the ovary showed a marked increase during ovarian maturation (*p* < 0.05) (Table 2). The protein contents in the muscle showed no significant change, but an increase in muscle lipid was found from Stage I to II and then remained constant (Table 2).

### 3.4. Amino Acid and the Relationship between Ovary Stages

To examine the associations between ovary developmental stages and the amino acid composition, a principal component analysis (PCA) was conducted. The first two components cumulatively explained 89.12%, 89.12%, and 89.26% of the variation in the free amino acids of the ovary, hepatopancreas, and muscle, respectively (Figure 3, Appendix A). In the ovary, the eigenvectors of amino acids for each PCA showed that only Trp and Glu positively contributed to PC2, while the remaining amino acids positively related to PC1, except for Phe (Figure 3A, Appendix A). The scores of ovary stages showed a significant increase from Stage I to V in PC1, while initially increasing from Stage I to IV and then decreasing in PC2 (Appendix A). In muscle, the eigenvectors of amino acids for each PCA showed that most essential amino acids positively contributed to PC1, while the remaining amino acids positively contributed to PC2 (Figure 3C, Appendix A).

During the process of ovary development from Stage I to V, the contents of Ile, Leu, and Val of essential amino acids (EAA) increased in ovaries and muscles, whereas the content of Val in the hepatopancreas decreased (Table 3). Although Trp contents in all three tissues were low, they increased in the hepatopancreas but decreased in the ovaries and muscles during maturation (Table 3). Additionally, the relative contents of umami amino acids (Glu and Asp) showed the highest concentrations in all three tissues. Moreover, the relative contents of sweet amino acids (Ala and Ser) increased during maturation, while Glu decreased, even though Glu was the dominant amino acid in all three tissues during all stages of ovarian maturation (Table 3).

### 3.5. Fatty Acid and the Relationship between Ovary Stages

The first two components of PCA analysis cumulatively explained 74.38%, 53.58%, and 57.12% of the variation in fatty acids of the ovary, hepatopancreas, and muscle, respectively (Figure 4, Appendix A). Most fatty acids in the ovary contributed to PC1, with saturated fatty acids (SFA) and PUFA showing positive correlations, while some MUFA (C24:1, C18:1, and C16:1) showed negative correlations with PC1 (Figure 4A, Appendix A). Furthermore, the PC scores of ovary stages strongly decreased from Stage I to V in PC1.

Among the various types of fatty acids analyzed in this study, SFA was found to be dominant in both the ovary and muscle tissues at all developmental stages, followed by MUFA and PUFA (Table 4). However, in the hepatopancreas tissue, the values of MUFA (mainly 18:1n-9 and 24:1n-9) were dominant during the ovary maturation process (Table 4). We observed an increasing trend in the levels of MUFA and HUFA in the ovary tissue during maturation, while the hepatopancreas and muscle tissues showed less pronounced changes (*p* < 0.05) (Table 4). The levels of n-6 HUFA, especially C20:4n-6 (ARA), increased in the ovary and muscle tissues from Stage I to III, and then decreased thereafter (*p* < 0.05) (Table 4).

### 3.6. Triglycerides, Cholesterol and Phospholipids

Triglyceride (TG) content in the ovary increased during the maturation process (*p* < 0.05) (Figure 5A), but the content was lower than that in the hepatopancreas. We found that TG did not show a significant change in the muscle during the entire ovary maturation process (*p* > 0.05) (Figure 6A). The cholesterol (CHOL) content in the ovary increased significantly from Stage I to IV (*p* < 0.05), and then slightly decreased at Stage V (*p* > 0.05) (Figure 5B). However, there were no significant changes in the CHOL content of the hepatopancreas and muscle during the ovary maturation. The total phospholipid (PL) content in the ovary displayed a significant increase across the different stages of ovarian development (*p* < 0.05), of which contents exceeded the contents of the hepatopancreas and muscle (Figure 5C). The total PL content in the muscle increased from Stage I to Stage IV but showed a significant decrease at Stage V (*p* < 0.05) (Figure 5C).

The PL compositions (phosphatidylinositol: PI; phosphatidylcholine: PC; phosphatidylethanolamine: PE; phosphatidylserine: PS; cardiolipins: CL) in the ovary were significantly higher than those observed in the hepatopancreas and muscle (Figure 6). Across all three tissues, the PC was dominant and showed a significant increase during the ovarian stages (*p* < 0.05).

### 3.7. Carotenoid Content

During the ovary maturation process, the carotenoid content in the ovary ranged from 2.84 to 3.53 mg/100 g tissue, which was lower than that of the hepatopancreas but higher than that of the muscle (Figure 7). Both the ovary and hepatopancreas showed a significant increase in carotenoid content from Stage I to Stage V (*p* < 0.05), while those of the muscle showed a sharp increase only at Stage V (*p* < 0.05) (Figure 7).

## 4. Discussion

### 4.1. GSI, HSI, and Ovary Maturation

*P. trituberculatus* ovarian maturation was classified into five distinct stages based on external morphology and ovary color. Similar classification systems have been reported for other species of female crabs such as *Scylla serrata* [33], *S. paramamosain* [34], and *S. olivacea* [35]. The GSI, which serves as an indicator of gonadal maturation in *P. trituberculatus* [36], showed a clear increase across the different ovary stages observed in the present study. In addition, the hepatopancreas digest and absorb nutrients from the diet, then transport the nutrients to the ovary via hemolymph during oogenesis [37]. The HSI is commonly employed as an indicator of the energy reserve status in the hepatopancreas [38]. We found a significant increasing trend in the hepatosomatic index (HSI) during ovarian maturation. Accordingly, HSI was positively related to GSI in *P. trituberculatus*, which is similar to those reported in the shrimp *Pleoticus muelleri* [39]. However, some studies have reported that the HSI of crustaceans remains unchanged [40] or even decreases [4,41] with ovarian development. Given that nutrient and energy storage in organisms are typically influenced by food availability, such as the abundance, quality, and portion size of food [42], the increased HSI during ovary maturation in the present study suggests that the female samples were collected from habitats with ample food supply, potentially promoting *P. trituberculatus* oogenesis.

### 4.2. Total Protein and Amino Acid Contents

The protein levels in the diet of aquatic bloodstocks may have significant effects on ovarian development and egg laying quality [43]. The rapid ovarian development process is characterized by an intensive period of protein synthesis [43,44]. In this study, we observed an increase in protein content in the ovary and a decrease in the hepatopancreas during ovarian maturation. Similar findings have been reported in other crustaceans, such as *Fenneropenaeus merguiensis* (Fatima et al., 2013) [45] and *Portunus pelagicus* [46], suggesting that proteins or their precursors may be transferred from the hepatopancreas to the ovary via hemolymph.

During ovary maturation, the synthesis of maturation-related substances, such as yolk protein, peptide hormones, enzymes, and egg formation, requires a large number of proteins that are derived from amino acids [44]. Additionally, amino acids are the primary source of energy for the embryo of most marine fish and crustaceans [43]. In this study, we found that amino acids exhibited a tissue-specific expression pattern. Among them, the contents of essential amino acids (EAAs) were positively correlated with ovary development stages, indicating the crucial role of EAA in ovarian maturation and subsequent embryonic and larval development [11,47]. Among the essential amino acids (EAAs), L-isoleucine (Ile), L-leucine (Leu), and L-valine (Val) belong to the branched-chain amino acids (BCAAs), which act as signal molecules that activate life processes ranging from protein synthesis to insulin secretion [48]. In this study, we found that the contents of Ile, Leu, and Val in the ovaries and muscles of *P. trituberculatus* increased from ovary Stage I to V, indicating the stage-specific requirements of these EAAs during ovarian maturation. In addition, the high BCAA contents in floating fish eggs [49] and in different developmental stages of crab *Eriocheir sinensis* embryos [50] suggest that BCAAs are critical for embryonic development. Moreover, BCAAs are widely believed to activate the mTOR cell signaling pathway, thereby initiating protein synthesis in skeletal muscle and the intestine [51,52]. The mTOR pathway regulates growth, molting, and cell differentiation in crustaceans [53,54], and more importantly, it regulates the expression of vitellogenin to control oocyte maturation, ovarian development, and fecundity in invertebrates [8]. Therefore, exogenous BCAAs are necessary during the maturation process of this species.

Moreover, tryptophan (Trp) is an essential amino acid with multiple metabolic functions, including its role in protein synthesis [55] and immune metabolism [56]. In this study, we found that Trp contents were relatively low in all three tissues compared to other essential amino acids, but its concentration increased in the hepatopancreas while decreasing in ovaries and muscles during the maturation process in *P. trituberculatus.* This discovery implies that Trp primarily functions in the hepatopancreas of crustaceans, which serves as a digestive organ of the immune system [37]. A previous study reported that over 95% of Trp in immune organs generate immune-related metabolites, such as enhancing antioxidant enzyme activity, which is regulated by the kynurenine pathway [55], and stimulating toll-like receptors, which serve as the first line of defense against pathogen invasion [57]. In the future, studies are necessary to confirm the immune function of exogenous Trp in the diet of *P. trituberculatus*.

Amino acids not only serve as the building blocks for reproductive protein synthesis but also play a direct role in flavor perception. We found that Glu and Asp, which are umami amino acids, were present in the highest concentrations in three different tissues of *P. trituberculatus*, consistent with previous studies on other crab species such as *S. paramamosain* [58], *E. sinensis* [59], and *Charybdis japonica* [60]. Furthermore, during ovary maturation, the relative content of sweet amino acids (Ala and Ser) increased, while umami taste amino acid (Glu) decreased, suggesting that the maturation process leads to a decrease in freshness but an increase in sweetness in the ovary of *P. trituberculatus*. Also, previous research on E. sinensis reported high Glu contents in different embryo developmental stages [50], suggesting a potential role of Glu in crab embryo development. Consequently, enhancing the dietary intake of flavor-related amino acids, particularly Asp and Glu, could potentially have positive effects on reproduction and the development of embryos and larvae in crustaceans.

### 4.3. Total Lipid, Lipid Profiles and Fatty Acid Contents

Lipids serve as a crucial energy source for numerous reproductive processes in marine animals, including spawning and development of embryos and larvae [10]. Furthermore, lipids (including phospholipids, sterols, and fatty acids) also serve as nutrients that are vital for oogenesis and the formation of eggs [61]. Similar to previous studies conducted on crustacean species [62,63], the lipids in the hepatopancreas and muscle did not significantly change during maturation. However, the total lipids in the ovary showed an increasing trend before the proliferative stage of *P. trituberculatus*. Our findings suggest that during ovary maturation, the reserved lipid may be preferentially allocated to the ovary to promote maturation, fecundity, and embryo development [62]. The lipid class displays tissue specificity during the ovary maturation process. Triglycerides (TGs) are the major constituent of total lipids and are stored in the adipose tissue of the ovary and hepatopancreas and serve as the primary energy source for body function [44]. We observed the increases in TG in the ovary and hepatopancreas during maturation, the pattern of which was similar to that of mammals during the oocyte maturation process [64]. This result is possibly due to the heightened energy requirements for maturation and maintenance of metabolic homeostasis [44]. Additionally, cholesterol (CHOL) plays a vital role in the formation of new tissues [65] and serves as a precursor for steroid hormones, including estradiol, which is critical for ovary maturation and reproductive processes in crustaceans [66]. Phospholipids (PLs) facilitate the transport of CHOL and TG from the hepatopancreas into the hemolymph and accumulate in the ovary for vitellogenesis [67,68], oogenesis, and larval development after spawning [67,69]. In this study, we observed that the CHOL and PL contents accumulated in the ovary from the pre-developmental stage to the pre-mature stage, while their contents changed slightly in the hepatopancreas. Our findings emphasize the importance of supplementing feed with TG, CHO, and PL during the ovary maturation of *P. trituberculatus.*

The fatty acid composition and content of aquatic products are indicators for assessing the nutritional value of animals at different life stages and evaluating the quality of ovarian development and egg production [12,13]. This study reveals increasing trends of MUFA and LC-PUFA in the ovary during maturation, while hepatopancreas and muscle tissues showed less obvious changes. LC-PUFA accumulated in the mature ovary, contributing to initial vitellogenesis, oogenesis [16,63], and spawning activities of both marine and freshwater animals [70,71]. In crustacean larvae, MUFA supports the synthesis of PL [72] and is prioritized as an energy source for embryonic and larval fish metabolism [73,74]. Our results highlight the importance of MUFA and LC-PUFA in the ovary, acting as energy reservoirs and essential components for ovarian development [75]. Notably, although n-6 PUFA (especially ARA and LA) increased in the ovary from the pre-developmental stage to the proliferative stage, it decreased from the proliferative stage to the mature stage of *P. trituberculatus*. This decrease may delay ovarian maturation [76], reduce fecundity, and negatively affect the egg quality and larval hatching rate [77]. ARA is accumulated in organisms either from food or by the desaturation and chain elongation of linoleic acid (LA, 18:2n-6) [78]. In egg yolk, ARA is preferentially incorporated into the structural lipids of larval tissue [15,79]. These results highlight the necessity of including ARA and LA in *P. trituberculatus* broodstock diets, and their subsequent contribution to the larvae.

### 4.4. Carotenoid Content

Carotenoids help to protect developing oocytes [34] and improve fecundity and spawning quality, such as the percentage of live eggs, hatching rate, and larval development [80,81]. Similar to other crustaceans [21,82], an increasing tendency of carotenoid content in the ovary and hepatopancreas from the pre-developmental stage to the proliferative stage was observed in our study. However, there was little change in the muscle during the maturation process. The free and esterified forms of carotenoids may initially accumulate in the hepatopancreas, then be mobilized to the hemolymph as carotenoglycolipoproteins, and subsequently sequestered to the ovaries, where they accumulate in the oocytes as part of the vitellin during the proliferative stage [19,20]. Our findings confirm the transportation of carotenoids among tissues during ovary maturation and highlight the potential role of carotenoids in facilitating the rapid maturation of oocytes.

Crustaceans cannot synthesize carotenoids de novo; hence, their diet is the exclusive source of carotenoid accumulation in the tissues of *P. trituberculatus.* Carotenoids are rich sources in marine plants and mollusks [83], which are frequently consumed by various crustaceans, including crabs [84]. Ghazali et al. (2017) [35] demonstrated that crustaceans being fed diets rich in carotenoids can undergo color changes in their tissues, transitioning from a light yellow to bright orange [35]. Therefore, the characteristic color changes in the ovary not only serve as a basis for evaluating the degree of maturation but also act as an indicator of food quality.

## 5. Conclusions

The process of ovarian maturation in *P. trituberculatus* can be classified into five stages based on the developmental characteristics of oocytes. Although the protein content remained unchanged in the ovary with the maturation process, most of the EAA in three tissues increased significantly during ovarian development, indicating the important role of EAA in oogenesis. The total lipid content of the ovary, as well as the lipid fractions, MUFA, LC-PUFA, and carotenoid, mainly accumulated in the ovary and increased with ovarian development. However, most of them in the hepatopancreas and/or muscle remained relatively stable. These findings suggest that reserved lipids may be preferentially allocated to the ovary to promote fecundity and embryo development. Overall, the changes in nutritional values among tissues during ovarian maturation suggest that the distribution and utilization of nutrients among different tissues in organisms are complex and refined and that there may be interactions among tissues. These findings help us better understand the nutritional metabolism and reproductive physiology of organisms. Furthermore, our findings provide valuable information for the development of a suitable broodstock diet for *P. trituberculatus.* In the future, it would be necessary to confirm the effect of the addition of these nutrients to the diet on the reproduction and larval development of *P. trituberculatus*.

## Figures and Tables

**Figure 1 animals-13-03220-f001:**
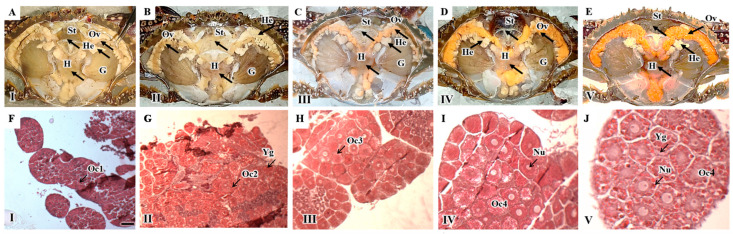
Inner morphology and histological observation of female *Portunus trituberculatus* at different developmental stages. (**A**,**F**): Pre-developmental stage; (**B**,**G**): Initial developmental stage; (**C**,**H**): Proliferative stage; (**D**,**I**): Pre-maturation stage; (**E**,**J**): Mature stage. Ov, ovary; He, hepatopancreas; H, heart; G, gill; St, stomach. Og, Oogonia; Oc1, Stage 1 oocyte; Oc2, Stage 2 oocytes; Oc3, Stage 3 oocyte; Oc4, Stage 4 oocytes; Nu, nucleus; Yg, yolk granules. Bar: 100 μm.

**Figure 2 animals-13-03220-f002:**
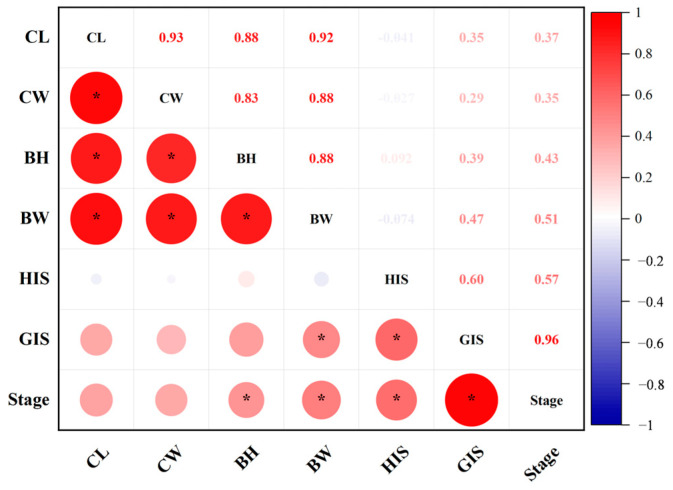
Correlation among GSI, his, and growth traits of female *Portunus trituberculatus*. GSI, gonadosomatic index; HIS, hepatosomatic index; CL, carapace length; CW, carapace width; BH, body height; BW, body weight. Red and blue numbers indicate the positive and negative correlation coefficients, respectively. * represents the significant difference *p* < 0.05.

**Figure 3 animals-13-03220-f003:**
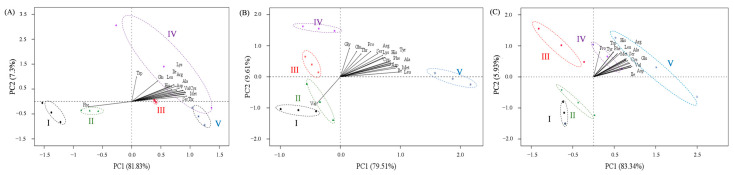
Results of PCA showing a biplot of PC scores of five gonad development stages (I, II, III, IV, and V) and loadings of amino acids in ovary (**A**), hepatopancreas (**B**), and muscle (**C**). (Asp, aspartate; Thr, threonine; Ser, serine; Glu, glutamate; Gly, glycine; Ala, alanine; Cys, cysteine; Val, valine; Met, methionine; Ile, isoleucine; Leu, leucine; Tyr, tyrosine; Phe, phenylalanine; His, histidine; Lys, lysine; Arg, arginine; Pro, proline; Try, tryptophan). Panels (**A**–**C**) represent biplots of the PC1 and PC2 in ovary, hepatopancreas, and muscle, respectively.

**Figure 4 animals-13-03220-f004:**
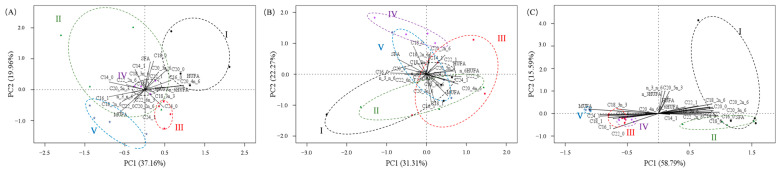
Results of PCA showing a biplot of PC scores of five gonad development stages (I, II, III, IV, and V) and loadings of fatty acids in ovary (**A**), hepatopancreas (**B**), and muscle (**C**). “_” represents “:” in each figure.

**Figure 5 animals-13-03220-f005:**
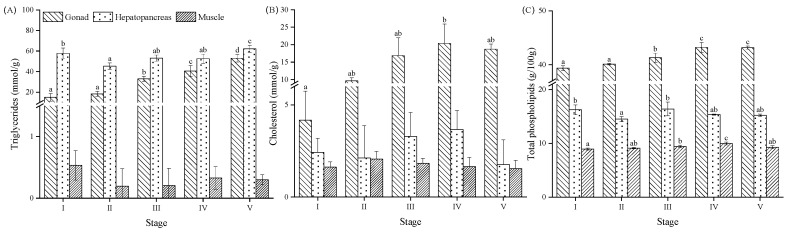
Triglyceride (**A**), cholesterol (**B**), and phospholipid (**C**) contents in gonad, hepatopancreas, hemolymph, and muscle of *Portunus trituberculatus* at different ovarian maturation stages. Data are presented as mean ± SD (*n* = 4). Different letters are statistically different between ovarian stages (*p* < 0.05).

**Figure 6 animals-13-03220-f006:**
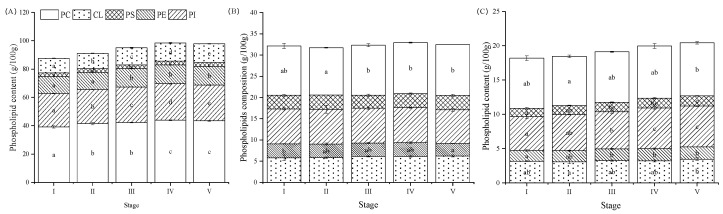
Phospholipid compositions (phosphatidylinositol, PI; phosphatidylcholine, PC; phosphatidylethanolamine, PE; phosphatidylserine, PS; cardiolipins, CL) in ovary (**A**), hepatopancreas (**B**) and muscle (**C**) of *Portunus trituberculatus* at different ovarian maturation stages. Data are presented as mean ± SD (*n* = 3). Different letters are statistically different between ovarian stages (*p* < 0.05).

**Figure 7 animals-13-03220-f007:**
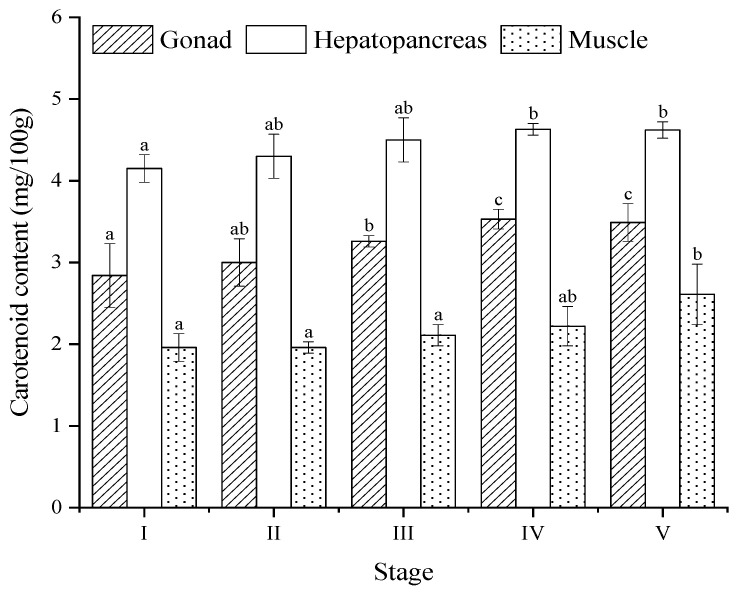
Total carotenoid content in ovary, hepatopancreas, and muscle of *Portunus trituberculatus* at different ovarian maturation stages. Data are presented as mean ± SD (*n* = 3). Different letters are statistically different (*p* < 0.05) between ovarian stages.

**Table 1 animals-13-03220-t001:** Comparison of gonadosomatic index, hepatosomatic index, and growth traits among different ovarian maturation stages of *Portunus trituberculatus*.

Stage	Stage I	Stage II	Stage III	Stage IV	Stage V
CL (mm)	78.53 ± 10.91 ^a^	85.12 ± 8.44 ^ab^	85.29 ± 4.17 ^ab^	87.82 ± 7.13 ^b^	87.32 ± 3.86 ^b^
CW (mm)	165.43 ± 8.73 ^a^	185.25 ± 15.15 ^b^	184 ± 11.34 ^b^	180.2 ± 11.41 ^b^	184.8 ± 8.61 ^b^
BH (mm)	38.91 ± 8.86	45.69 ± 4.44	43.72 ± 1.77	45.11 ± 2.46	44.28 ± 0.6
BW (g)	260.26 ± 37.65 ^a^	321.29 ± 38.99 ^b^	378.84 ± 34.39 ^b^	361.37 ± 52.71 ^b^	361.38 ± 32.98 ^b^
HSI (%)	5.39 ± 0.58 ^a^	6.17 ± 1.64 ^ab^	7.16 ± 1.31 ^b^	7.52 ± 1 ^b^	7.46 ± 1.73 ^b^
GSI (%)	0.48 ± 0.17 ^a^	0.7 ± 0.08 ^ab^	1.77 ± 0.53 ^ab^	2.29 ± 0.16 ^b^	4.22 ± 1.08 ^b^

CL, carapace length; CW, carapace width; BH, body height; BW, body weight; HSI, hepatosomatic index; GSI, gonadosomatic index. Data are presented as mean ± SD (*n* = 4). Different letters are statistically different between ovarian stages (*p* < 0.05).

**Table 2 animals-13-03220-t002:** The proximate composition (%) of muscle, hepatopancreas, and ovary of female *Portunus trituberculatus* during ovarian development.

		Stage I	Stage II	Stage III	Stage IV	Stage V
Muscle	Moisture (%)	80.66 ± 1.1 ^bc^	81.32 ± 0.64 ^b^	79.19 ± 0.25 ^ab^	79.24 ± 1.16 ^ab^	79.78 ± 2.44 ^a^
	Protein (%)	84.38 ± 2.28	83.01 ± 0.56	82.67 ± 1.91	82.95 ± 0.88	82.77 ± 1.98
	Lipid (g/100 g)	1.39 ± 0.03 ^a^	1.5 ± 0.07 ^b^	1.54 ± 0.03 ^b^	1.53 ± 0.06 ^b^	1.5 ± 0.07 ^b^
Hepatopancreas	Moisture (%)	70.98 ± 2.86 ^b^	70.15 ± 3.39 ^b^	66.58 ± 0.6 ^b^	61.03 ± 1.85 ^a^	60.58 ± 4.8 ^a^
	Protein (%)	39.17 ± 5.94 ^c^	30 ± 4.19 ^ab^	32.51 ± 0.88 ^b^	26.01 ± 2.02 ^a^	27.57 ± 2.53 ^ab^
	Lipid (g/100 g)	2.84 ± 0.06	2.62 ± 0.13	2.82 ± 0.17	2.64 ± 0.08	2.67 ± 0.11
Gonad	Moisture (%)	78.28 ± 0.43 ^c^	74.45 ± 5.27 ^c^	61.95 ± 2.42 ^b^	62.43 ± 1.36 ^b^	57.88 ± 1.76 ^a^
	Protein (%)	57.34 ± 6.79 ^a^	56.48 ± 2.41 ^a^	65.88 ± 1.87 ^b^	66.54 ± 1.1 ^b^	66.92 ± 1.86 ^b^
	Lipid (g/100 g)	3.7 ± 0.09 ^a^	3.9 ± 0.09 ^b^	4.13 ± 0.11 ^c^	4.33 ± 0.12 ^d^	4.31 ± 0.04 ^d^

Values are presented as means ± SD (*n* = 4). Mean values in the same row with different superscript letters are significantly different (*p* < 0.05).

**Table 3 animals-13-03220-t003:** Amino acids composition (% total amino acids) of female *Portunus trituberculatus* ovary during ovarian development.

Tissues	Amino Acids	Stage I	Stage II	Stage III	Stage IV	Stage V
Gonad	Asp	8.16 ± 0.71 ^y^	8.51 ± 0.35 ^y^	8.62 ± 0.03 ^y^	8.67 ± 0.01 ^y^	8.41 ± 0.09 ^x^
Hepatopancreas		8.97 ± 0.01 ^by^	8.69 ± 0.07 ^ay^	8.64 ± 0.02 ^ay^	8.8 ± 0.03 ^aby^	8.92 ± 0.19 ^bx^
Muscle		7.22 ± 0.04 ^ax^	7.22 ± 0.05 ^ax^	7.24 ± 0.02 ^ax^	7.63 ± 0.26 ^bx^	8.19 ± 0.24 ^cy^
Gonad	Thr *	4.53 ± 0.06 ^aby^	4.48 ± 0.05 ^aby^	4.6 ± 0.06 ^by^	4.41 ± 0.14 ^ay^	4.57 ± 0.05 ^aby^
Hepatopancreas		4.87 ± 0.01 ^cz^	4.88 ± 0.03 ^cz^	4.89 ± 0.05 ^cz^	4.73 ± 0.06 ^bz^	4.57 ± 0.03 ^ax^
Muscle		3.15 ± 0.02 ^abcx^	3.07 ± 0.01 ^ax^	3.20 ± 0.10 ^bcx^	3.23 ± 0.01 ^cx^	3.10 ± 0.07 ^aby^
Gonad	Ser	4.98 ± 0.05 ^ay^	5.01 ± 0.08 ^aby^	5.18 ± 0.01 ^by^	5.74 ± 0.21 ^cy^	6.02 ± 0.06 ^dy^
Hepatopancreas		4.66 ± 0.02 ^bx^	4.66 ± 0.03 ^bx^	4.65 ± 0.03 ^bx^	4.61 ± 0.02 ^bx^	4.47 ± 0.06 ^ay^
Muscle		6.02 ± 0.15 ^abz^	5.98 ± 0.01 ^az^	5.97 ± 0.05 ^az^	6.16 ± 0.13 ^bz^	6.06 ± 0.01 ^abx^
Gonad	Glu	10.15 ± 0.39 ^cx^	10.07 ± 0.1 ^bcx^	9.67 ± 0.18 ^abx^	9.6 ± 0.21 ^ax^	9.46 ± 0.06 ^ax^
Hepatopancreas		10.36 ± 0.11 ^dx^	10.08 ± 0.14 ^bcx^	10.21 ± 0.03 ^cdy^	9.98 ± 0.07 ^by^	9.61 ± 0.10 ^ay^
Muscle		13.56 ± 0.01 ^by^	13.36 ± 0.2 ^aby^	13.23 ± 0.24 ^az^	13.11 ± 0.12 ^az^	13.09 ± 0.07 ^ax^
Gonad	Pro	4.07 ± 0.02 ^ay^	4.14 ± 0.02 ^az^	4.10 ± 0.06 ^ay^	4.41 ± 0.15 ^bz^	4.51 ± 0.12 ^bz^
Hepatopancreas		4.08 ± 0.04 ^abcy^	4.06 ± 0.01 ^aby^	4.12 ± 0.03 ^bcy^	4.18 ± 0.11 ^cy^	3.99 ± 0.01 ^ax^
Muscle		3.14 ± 0.04 ^bx^	3.03 ± 0.05 ^abx^	3.15 ± 0.16 ^bx^	2.95 ± 0.03 ^ax^	2.92 ± 0.08 ^ay^
Gonad	Gly	9.57 ± 0.02 ^cx^	9.66 ± 0.02 ^dx^	9.51 ± 0.05 ^cx^	9.18 ± 0.04 ^ax^	9.35 ± 0.06 ^bx^
Hepatopancreas		10.82 ± 0.03 ^by^	10.73 ± 0.12 ^by^	10.71 ± 0.08 ^by^	11.02 ± 0.07 ^cy^	10.17 ± 0.14 ^az^
Muscle		11.74 ± 0.08 ^z^	11.80 ± 0.03 ^z^	11.63 ± 0.36 ^z^	11.97 ± 0.02 ^z^	11.91 ± 0.18 ^y^
Gonad	Ala	5.01 ± 0.01 ^ay^	4.94 ± 0.03 ^ay^	5.15 ± 0.02 ^by^	5.13 ± 0.07 ^by^	4.97 ± 0.04 ^ay^
Hepatopancreas		5.28 ± 0.04 ^az^	5.30 ± 0.01 ^az^	5.29 ± 0.03 ^az^	5.32 ± 0.02 ^az^	5.48 ± 0.02 ^bx^
Muscle		4.78 ± 0.08 ^cx^	4.85 ± 0.01 ^dx^	4.76 ± 0.04 ^cx^	4.67 ± 0.01 ^bx^	4.53 ± 0.02 ^az^
Gonad	Cys	3.36 ± 0.01 ^az^	3.44 ± 0.02 ^bz^	3.70 ± 0.05 ^bz^	3.68 ± 0.07 ^by^	3.74 ± 0.01 ^bz^
Hepatopancreas		2.91 ± 0.06 ^bcx^	2.82 ± 0.05 ^ax^	2.84 ± 0.02 ^abx^	2.93 ± 0.03 ^cx^	2.95 ± 0.05 ^cy^
Muscle		3.17 ± 0.01 ^by^	3.17 ± 0.01 ^by^	3.15 ± 0.06 ^by^	3.09 ± 0.03 ^az^	3.05 ± 0.02 ^ax^
Gonad	Val *	5.62 ± 0.05 ^a^	5.91 ± 0.01 ^bx^	6.52 ± 0.11 ^cz^	6.55 ± 0.06 ^cz^	6.8 ± 0.04 ^dz^
Hepatopancreas		5.60 ± 0.06 ^c^	5.46 ± 0.07 ^cy^	5.09 ± 0.28 ^bx^	4.62 ± 0.05 ^ax^	4.61 ± 0.01 ^ay^
Muscle		5.57 ± 0.04 ^a^	5.54 ± 0.01 ^ay^	5.53 ± 0.14 ^ay^	5.58 ± 0.17 ^ay^	5.82 ± 0.26 ^bx^
Gonad	Met *	1.68 ± 0.03 ^ax^	1.98 ± 0.02 ^cy^	1.85 ± 0.03 ^bcy^	1.61 ± 0.05 ^abx^	1.74 ± 0.01 ^bcx^
Hepatopancreas		1.63 ± 0.04 ^ax^	1.63 ± 0.02 ^ax^	1.69 ± 0.01 ^bx^	1.75 ± 0.01 ^cx^	2.04 ± 0.04 ^dz^
Muscle		2.04 ± 0.05 ^ay^	2.28 ± 0.01 ^bz^	2.19 ± 0.04 ^abz^	2.20 ± 0.04 ^by^	2.18 ± 0.01 ^aby^
Gonad	Ile *	4.40 ± 0.01 ^az^	4.57 ± 0.01 ^by^	4.58 ± 0.03 ^bz^	4.60 ± 0.14 ^bz^	4.57 ± 0.01 ^bz^
Hepatopancreas		4.23 ± 0.02 ^ax^	4.24 ± 0.05 ^ax^	4.30 ± 0.01 ^bx^	4.37 ± 0.07 ^bx^	4.39 ± 0.02 ^by^
Muscle		4.37 ± 0.01 ^ay^	4.36 ± 0.01 ^ay^	4.47 ± 0.01 ^by^	4.46 ± 0.02 ^by^	4.54 ± 0.01 ^cx^
Gonad	Leu *	7.08 ± 0.14 ^ax^	7.13 ± 0.07 ^ax^	7.15 ± 0.07 ^abx^	7.21 ± 0.02 ^by^	7.27 ± 0.11 ^cx^
Hepatopancreas		7.11 ± 0.23 ^ax^	7.16 ± 0.01 ^axy^	7.15 ± 0.09 ^ax^	6.97 ± 0.04 ^ax^	7.9 ± 0.01 ^by^
Muscle		7.85 ± 0.02 ^ay^	7.85 ± 0.01 ^ay^	8.15 ± 0.06 ^by^	8.17 ± 0.02 ^bz^	8.11 ± 0.25 ^bz^
Gonad	Tyr	4.42 ± 0.02 ^cz^	4.14 ± 0.01 ^bcz^	4.07 ± 0.04 ^bz^	3.98 ± 0.01 ^az^	4.10 ± 0.02 ^bcz^
Hepatopancreas		3.52 ± 0.01 ^y^	3.48 ± 0.02 ^y^	3.49 ± 0.01 ^y^	3.43 ± 0.05 ^y^	3.45 ± 0.06 ^x^
Muscle		2.84 ± 0.02 ^bx^	2.82 ± 0.05 ^bx^	2.81 ± 0.01 ^bx^	2.49 ± 0.02 ^ax^	2.44 ± 0.07 ^ay^
Gonad	Phe *	4.98 ± 0.03 ^cy^	4.23 ± 0.73 ^b^	3.56 ± 0.04 ^ax^	3.49 ± 0.01 ^ax^	3.37 ± 0.02 ^ax^
Hepatopancreas		4.63 ± 0.01 ^cx^	4.5 ± 0.01 ^bc^	4.51 ± 0.01 ^cy^	4.30 ± 0.05 ^ay^	4.33 ± 0.07 ^aby^
Muscle		4.99 ± 0.08 ^cz^	4.90 ± 0.02 ^c^	4.84 ± 0.02 ^bcz^	4.61 ± 0.01 ^bz^	4.55 ± 0.13 ^ay^
Gonad	Lys *	8.69 ± 0.20 ^by^	8.71 ± 0.03 ^by^	8.53 ± 0.05 ^ab^	8.52 ± 0.09 ^ab^	8.23 ± 0.01 ^ax^
Hepatopancreas		8.03 ± 0.12 ^ax^	8.44 ± 0.1 ^bx^	8.56 ± 0.14 ^b^	8.44 ± 0.20 ^b^	8.53 ± 0.04 ^bx^
Muscle		8.71 ± 0.03 ^by^	8.79 ± 0.01 ^by^	8.64 ± 0.05 ^b^	8.38 ± 0.02 ^a^	8.43 ± 0.07 ^ay^
Gonad	His *	2.38 ± 0.06 ^bz^	2.37 ± 0.02 ^bz^	2.40 ± 0.05 ^bz^	2.25 ± 0.02 ^az^	2.27 ± 0.06 ^az^
Hepatopancreas		1.88 ± 0.07 ^ay^	1.97 ± 0.01 ^by^	1.99 ± 0.02 ^by^	2.00 ± 0.01 ^by^	2.00 ± 0.01 ^bx^
Muscle		1.35 ± 0.01 ^cx^	1.34 ± 0.01 ^cx^	1.36 ± 0.01 ^cx^	1.31 ± 0.02 ^bx^	1.27 ± 0.01 ^ay^
Gonad	Arg	8.44 ± 0.13 ^aby^	8.31 ± 0.07 ^ax^	8.49 ± 0.16 ^abx^	8.62 ± 0.05 ^bx^	8.38 ± 0.04 ^ax^
Hepatopancreas		8.73 ± 0.01 ^az^	9.05 ± 0.09 ^by^	9.13 ± 0.07 ^by^	9.74 ± 0.11 ^cy^	9.68 ± 0.05 ^cy^
Muscle		8.15 ± 0.03 ^ax^	8.31 ± 0.17 ^abx^	8.36 ± 0.08 ^bx^	8.64 ± 0.12 ^cx^	8.54 ± 0.03 ^cz^
Gonad	Trp *	2.46 ± 0.02 ^by^	2.42 ± 0.02 ^by^	2.33 ± 0.02 ^aby^	2.35 ± 0.17 ^aby^	2.22 ± 0.02 ^ay^
Hepatopancreas		2.69 ± 0.05 ^az^	2.86 ± 0.09 ^bcz^	2.75 ± 0.08 ^abz^	2.83 ± 0.07 ^bcz^	2.91 ± 0.04 ^cx^
Muscle		2.46 ± 0.02 ^bx^	2.42 ± 0.02 ^bx^	2.33 ± 0.02 ^abx^	2.35 ± 0.17 ^abx^	2.22 ± 0.02 ^az^

Asp, aspartate; Thr, threonine; Ser, serine; Glu, glutamate; Gly, glycine; Ala, alanine; Cys, cysteine; Val, valine; Met, methionine; Ile, isoleucine; Leu, leucine; Tyr, tyrosine; Phe, phenylalanine; His, histidine; Lys, lysine; Arg, arginine; Pro, proline; Try, tryptophan; Essential amino acids are marked with *. Values are presented as means ± SD (*n* = 3). Different superscript letters in a row and columns indicate significant differences among stages and tissues, respectively (*p* < 0.05).

**Table 4 animals-13-03220-t004:** Fatty acid composition (% total fatty acid) of female *Portunus trituberculatus* ovaries during ovarian development.

Tissues	Fatty Acid	Stage I	Stage II	Stage III	Stage IV	Stage V
Gonad	C14:0	3.25 ± 0.48 ^dy^	2.77 ± 0.13 ^cy^	1.57 ± 0.12 ^ax^	2.21 ± 0.16 ^by^	2.05 ± 0.08 ^bx^
Hepatopancreas		6.26 ± 1.58 ^z^	6.11 ± 1 ^z^	6.28 ± 0.33 ^y^	6.63 ± 0.45 ^z^	6.45 ± 0.83 ^y^
Muscle		1.56 ± 0.24 ^x^	1.63 ± 0.22 ^x^	2.39 ± 0.7 ^x^	1.37 ± 0.16 ^x^	1.52 ± 0.14 ^x^
Gonad	C16:0	32.38 ± 1.77 ^cy^	32.49 ± 1.11 ^cy^	21.56 ± 1.21 ^ax^	21.98 ± 1.02 ^by^	20.09 ± 0.51 ^ay^
Hepatopancreas		4.16 ± 1.05 ^cx^	2.63 ± 0.91 ^abx^	3.63 ± 0.17 ^bcy^	2.94 ± 0.47 ^abx^	2.22 ± 0.11 ^ax^
Muscle		34.05 ± 6.37 ^y^	33.66 ± 5.59 ^y^	30.93 ± 1.34 ^z^	34.61 ± 2.42 ^z^	30.43 ± 3.06 ^z^
Gonad	C18:0	14.62 ± 2.14 ^bx^	16.57 ± 0.86 ^cxy^	10.75 ± 0.65 ^ax^	9.59 ± 0.29 ^ax^	9.1 ± 0.2 ^ax^
Hepatopancreas		16.19 ± 0.6 ^cx^	15.03 ± 0.91 ^bx^	14.52 ± 0.32 ^aby^	14.7 ± 0.83 ^aby^	13.72 ± 0.53 ^cy^
Muscle		18.66 ± 0.63 ^ay^	17.36 ± 1.67 ^ay^	18.16 ± 1.28 ^az^	22.41 ± 0.9 ^bz^	19.23 ± 2.65 ^az^
Gonad	C20:0	2.95 ± 0.78 ^b^	3.39 ± 0.19 ^by^	0.99 ± 0.12 ^abx^	0.75 ± 0.03 ^abx^	0.61 ± 0.02 ^ax^
Hepatopancreas		1.89 ± 0.19 ^c^	1.68 ± 0.22 ^bcx^	1.51 ± 0.04 ^abx^	1.49 ± 0.11 ^aby^	1.31 ± 0.18 ^ay^
Muscle		1.77 ± 0.77	1.7 ± 0.04 ^x^	2.7 ± 0.58 ^y^	3.08 ± 0.59 ^z^	2.73 ± 0.27 ^z^
Gonad	C22:0	4.6 ± 2.99 ^ax^	6.71 ± 0.33 ^ax^	9.19 ± 0.39 ^abxy^	9.17 ± 0.28 ^ab^	9.8 ± 0.22 ^b^
Hepatopancreas		8.02 ± 0.29 ^axy^	9.27 ± 0.75 ^ay^	8.53 ± 0.3 ^ax^	9.14 ± 0.2 ^a^	9.71 ± 0.75 ^b^
Muscle		9.93 ± 1.19 ^aby^	11.44 ± 1.38 ^by^	10.27 ± 1.41 ^aby^	8.31 ± 0.85 ^a^	9.71 ± 1.08 ^ab^
Gonad	C24:0	1.57 ± 0.29 ^ay^	1.33 ± 0.18 ^ay^	3.64 ± 0.18 ^dz^	3.22 ± 0.23 ^cy^	2 ± 0.09 ^by^
Hepatopancreas		2.66 ± 0.24 ^az^	2.9 ± 0.37 ^abz^	3.23 ± 0.15 ^by^	2.97 ± 0.19 ^aby^	2.85 ± 0.11 ^abz^
Muscle		0.53 ± 0.24 ^ax^	0.53 ± 0.28 ^ax^	0.78 ± 0.18 ^abx^	0.7 ± 0.15 ^abx^	0.95 ± 0.24 ^bx^
Gonad	SFA	59.38 ± 4.57 ^by^	63.27 ± 1.43 ^by^	47.71 ± 1.24 ^aby^	46.92 ± 0.93 ^aby^	43.65 ± 0.41 ^ay^
Hepatopancreas		39.18 ± 0.9 ^bx^	37.62 ± 1.38 ^abx^	37.7 ± 0.52 ^abx^	37.88 ± 1.01 ^abx^	36.28 ± 0.78 ^ax^
Muscle		66.49 ± 6.29 ^y^	65.14 ± 4.47 ^y^	64.61 ± 4.4 ^z^	70.2 ± 3.56 ^z^	64.57 ± 4.51 ^z^
Gonad	C14:1	0.14 ± 0.06 ^ax^	0.37 ± 0.25 ^b^	0.19 ± 0.02 ^abx^	0.15 ± 0 ^abx^	0.14 ± 0.02 ^ax^
Hepatopancreas		0.59 ± 0.11 ^by^	0.49 ± 0.06 ^ab^	0.5 ± 0.03 ^aby^	0.51 ± 0.03 ^aby^	0.47 ± 0.09 ^ay^
Muscle		0.49 ± 0.21 ^y^	0.57 ± 0.17	0.7 ± 0.1 ^z^	0.78 ± 0.25 ^z^	0.78 ± 0.23 ^z^
Gonad	C16:1	6.2 ± 1.46 ^aby^	5.15 ± 0.39 ^a^	7.18 ± 0.5 ^by^	9.33 ± 0.54 ^cy^	8.98 ± 0.22 ^cy^
Hepatopancreas		11.38 ± 1.44 ^az^	13.08 ± 1.33 ^ab^	12.33 ± 0.61 ^abz^	13.33 ± 0.88 ^bz^	13.94 ± 0.72 ^bz^
Muscle		3.83 ± 0.31 ^bx^	4.35 ± 0.31 ^b^	3.87 ± 0.81 ^bx^	2.96 ± 0.43 ^ax^	3.94 ± 0.59 ^bx^
Gonad	C18:1	13.53 ± 1.41 ^ax^	12.03 ± 0.77 ^ax^	16.98 ± 0.32 ^by^	16.21 ± 0.47 ^aby^	15.96 ± 0.13 ^aby^
Hepatopancreas		22.7 ± 0.26 ^by^	21.49 ± 1.3 ^aby^	21.38 ± 0.75 ^abz^	20.52 ± 1.03 ^bz^	21.35 ± 0.2 ^abz^
Muscle		12.33 ± 0.84 ^bx^	11.65 ± 1.33 ^bx^	10.06 ± 1.8 ^abx^	7.83 ± 2.06 ^ax^	9.95 ± 1.68 ^abx^
Gonad	C22:1	0.3 ± 0.05 ^b^	0.25 ± 0.1 ^ab^	0.18 ± 0.05 ^ax^	0.22 ± 0.03 ^abx^	0.18 ± 0.02 ^ax^
Hepatopancreas		0.43 ± 0.01 ^b^	0.32 ± 0.06 ^a^	0.33 ± 0.09 ^abx^	0.43 ± 0.03 ^by^	0.4 ± 0.02 ^aby^
Muscle		0.41 ± 0.23 ^ab^	0.31 ± 0.15 ^a^	0.6 ± 0.2 ^by^	0.53 ± 0.07 ^abz^	0.47 ± 0.05 ^abz^
Gonad	C24:1	10.53 ± 1.2 ^ax^	10.74 ± 0.59 ^ax^	17.39 ± 0.82 ^aby^	19.68 ± 1.73 ^aby^	24.57 ± 0.64 ^bz^
Hepatopancreas		15.27 ± 1.06 ^ay^	17.35 ± 1.38 ^by^	17.85 ± 0.37 ^by^	17.81 ± 0.7 ^by^	18.51 ± 1.18 ^by^
Muscle		9.11 ± 2.49 ^x^	9.65 ± 2.94 ^x^	9.47 ± 1.78 ^x^	8.69 ± 1.64 ^x^	10.61 ± 1.86 ^x^
Gonad	MUFA	30.7 ± 3.36 ^ax^	28.54 ± 1.28 ^ax^	41.92 ± 0.95 ^aby^	45.6 ± 0.91 ^by^	49.83 ± 0.3 ^by^
Hepatopancreas		50.37 ± 1.35 ^ay^	52.73 ± 1.44 ^by^	52.39 ± 0.39 ^bz^	52.6 ± 1.47 ^bz^	54.66 ± 0.31 ^cz^
Muscle		26.18 ± 3.44 ^x^	26.54 ± 3.75 ^x^	24.71 ± 4.09 ^x^	22.79 ± 3.39 ^x^	25.75 ± 3.72 ^x^
Gonad	C18:2n-6	0.3 ± 0.08 ^by^	0.26 ± 0.03 ^by^	0.06 ± 0.01 ^ax^	0.05 ± 0 ^ax^	0.06 ± 0 ^ax^
Hepatopancreas		0.05 ± 0.02 ^ax^	0.09 ± 0.05 ^abx^	0.04 ± 0.02 ^ax^	0.08 ± 0.07 ^ax^	0.16 ± 0.05 ^bx^
Muscle		0.28 ± 0.15 ^aby^	0.21 ± 0.07 ^ay^	0.53 ± 0.33 ^aby^	0.64 ± 0.24 ^by^	0.64 ± 0.28 ^by^
Gonad	C18:3n-6	0.09 ± 0.04 ^a^	0.14 ± 0.05 ^b^	0.05 ± 0.01 ^ax^	0.07 ± 0 ^ax^	0.07 ± 0.01 ^ax^
Hepatopancreas		0.2 ± 0.1	0.16 ± 0.06	0.2 ± 0.1 ^x^	0.13 ± 0.02 ^x^	0.11 ± 0.04 ^x^
Muscle		0.34 ± 0.3 ^a^	0.32 ± 0.2 ^a^	1.13 ± 0.69 ^cy^	0.46 ± 0.31 ^aby^	1.03 ± 0.17 ^bcy^
Gonad	C20:2n-6	0.8 ± 0.1 ^bxy^	1 ± 0.11 ^cy^	0.55 ± 0.08 ^ax^	0.48 ± 0.05 ^ax^	0.5 ± 0.04 ^ax^
Hepatopancreas		0.96 ± 0.06 ^ay^	1.01 ± 0.02 ^aby^	1.01 ± 0.03 ^aby^	0.97 ± 0.03 ^ay^	1.04 ± 0 ^by^
Muscle		0.61 ± 0.23 ^ax^	0.65 ± 0.28 ^ax^	1.03 ± 0.15 ^aby^	1.6 ± 0.5 c^z^	1.32 ± 0.33 ^bcy^
Gonad	C20:3n-6	0.4 ± 0.12 ^bz^	0.47 ± 0.12 ^by^	0.05 ± 0.01 ^ax^	0.03 ± 0.01 ^ax^	0.01 ± 0 ^ax^
Hepatopancreas		0.05 ± 0.02 ^bx^	0.05 ± 0.02 ^bx^	0.03 ± 0.01 ^abx^	0.02 ± 0 ^ax^	0.02 ± 0.01 ^ax^
Muscle		0.26 ± 0.05 ^y^	0.39 ± 0.11 ^y^	0.36 ± 0.18 ^y^	0.25 ± 0.04 ^y^	0.27 ± 0.07 ^y^
Gonad	C20:4n-6	6.29 ± 0.68 ^bcx^	5.71 ± 0.34 ^abx^	9.38 ± 0.33 ^dz^	6.56 ± 0.18 ^cy^	5.11 ± 0.25 ^ax^
Hepatopancreas		8.53 ± 1.79 ^by^	7.66 ± 0.59 ^ay^	8.05 ± 0.1 ^aby^	7.75 ± 0.63 ^abz^	7.02 ± 0.53 ^ay^
Muscle		5.09 ± 1.21 ^abx^	5.37 ± 0.41 ^abx^	5.96 ± 0.64 ^bx^	4.29 ± 0.62 ^ax^	4.56 ± 0.84 ^ax^
Gonad	C22:2n-6	0.1 ± 0.04 ^bx^	0.22 ± 0.02 ^c^	0.04 ± 0.01 ^ax^	0.05 ± 0.01 ^ax^	0.06 ± 0.01 ^ax^
Hepatopancreas		0.26 ± 0.07 ^by^	0.17 ± 0.02 ^a^	0.19 ± 0.05 ^abxy^	0.25 ± 0.04 ^by^	0.2 ± 0.03 ^aby^
Muscle		0.35 ± 0.12 ^aby^	0.27 ± 0.12 ^a^	0.36 ± 0.22 ^aby^	0.4 ± 0.06 ^abz^	0.56 ± 0.12 ^bz^
Gonad	C18:3n-3	0.16 ± 0.07 ^ax^	0.13 ± 0.03 ^ax^	0.19 ± 0.08 ^ax^	0.2 ± 0.05 ^ax^	0.63 ± 0.02 ^by^
Hepatopancreas		0.23 ± 0.14 ^x^	0.3 ± 0.04 ^y^	0.25 ± 0.1 ^x^	0.18 ± 0.04 ^x^	0.30 ± 0.07 ^x^
Muscle		0.54 ± 0.21 ^y^	0.42 ± 0.17 ^y^	0.64 ± 0.32 ^y^	0.74 ± 0.35 ^y^	0.56 ± 0.18 ^y^
Gonad	C20:5n-3	0.07 ± 0.01 ^x^	0.14 ± 0.04	0.02 ± 0 ^x^	0.02 ± 0 ^x^	0.05 ± 0 ^x^
Hepatopancreas		0.14 ± 0.04 ^abxy^	0.18 ± 0.04 ^b^	0.12 ± 0.03 ^axy^	0.11 ± 0.03 ^ay^	0.14 ± 0.03 ^aby^
Muscle		0.23 ± 0.1 ^y^	0.24 ± 0.09	0.26 ± 0.17 ^y^	0.30 ± 0.06 ^z^	0.27 ± 0.04 ^z^
Gonad	C22:6n-3	0.09 ± 0.03 ^x^	0.11 ± 0.02 ^x^	0.03 ± 0 ^x^	0.02 ± 0 ^x^	0.02 ± 0.01 ^x^
Hepatopancreas		0.04 ± 0.02 ^abx^	0.04 ± 0.01 ^abx^	0.02 ± 0.01 ^ax^	0.03 ± 0.01 ^abx^	0.07 ± 0.04 ^bx^
Muscle		0.54 ± 0.35 ^y^	0.28 ± 0.15 ^y^	0.4 ± 0.39 ^y^	0.34 ± 0.1 ^y^	0.48 ± 0.15 ^y^
Gonad	LC-PUFA	7.85 ± 0.57 ^ax^	8.18 ± 0.32 ^abx^	10.37 ± 0.48 ^b^	7.47 ± 0.25 ^abx^	6.52 ± 0.29 ^ax^
Hepatopancreas		10.45 ± 1.78 ^y^	9.65 ± 0.62 ^y^	9.91 ± 0.23	9.52 ± 0.47 ^y^	9.06 ± 0.56 ^y^
Muscle		8.24 ± 1.18 ^axy^	8.14 ± 0.75 ^ax^	10.68 ± 0.71 ^c^	9.01 ± 0.58 ^aby^	9.68 ± 0.91 ^bcy^
Gonad	n-6PUFA	7.98 ± 0.94 ^bxy^	7.8 ± 0.28 ^bx^	10.14 ± 0.4 ^c^	7.24 ± 0.22 ^bx^	5.82 ± 0.3 ^ax^
Hepatopancreas		10.05 ± 1.75 ^by^	9.14 ± 0.6 ^ay^	9.53 ± 0.12 ^ab^	9.2 ± 0.54 ^aby^	8.55 ± 0.49 ^ay^
Muscle		6.93 ± 1.37 ^ax^	7.2 ± 0.73 ^ax^	9.37 ± 0.83 ^b^	7.63 ± 0.8 ^ax^	8.37 ± 0.81 ^aby^
Gonad	n-3PUFA	0.28 ± 0.06 ^ax^	0.38 ± 0.08 ^ax^	0.24 ± 0.08 ^ax^	0.24 ± 0.05 ^ax^	0.7 ± 0.02 ^bx^
Hepatopancreas		0.4 ± 0.16 ^abx^	0.51 ± 0.06 ^bx^	0.38 ± 0.11 ^abx^	0.32 ± 0.08 ^ax^	0.51 ± 0.09 ^bx^
Muscle		1.31 ± 0.53 ^y^	0.94 ± 0.31 ^y^	1.31 ± 0.8 ^y^	1.38 ± 0.35 ^y^	1.31 ± 0.24 ^y^
Gonad	n-3/n-6	0.22 ± 0.36 ^b^	0.05 ± 0.01 ^ax^	0.02 ± 0.01 ^ax^	0.03 ± 0.01 ^ax^	0.12 ± 0.01 ^aby^
Hepatopancreas		0.04 ± 0.02 ^a^	0.06 ± 0.01 ^bx^	0.04 ± 0.01 ^ax^	0.04 ± 0.01 ^ax^	0.06 ± 0.01 ^bx^
Muscle		0.2 ± 0.12	0.13 ± 0.05 ^y^	0.14 ± 0.1 ^y^	0.19 ± 0.07 ^y^	0.16 ± 0.03 ^z^

Values are presented as means ± SD (*n* = 3). Different superscript letters in a row and columns indicate significant differences among stages and tissues, respectively (*p* < 0.05).

## Data Availability

Data generated or analyzed during this study are available in the Figshare Repository, https://doi.org/10.6084/m9.figshare.23565960.v1 (accessed on 23 June 2023).

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
