# Peer review of "Nutrient Composition of Ovary, Hepatopancreas and Muscle Tissues in Relation to Ovarian Development Stage of Female Swimming Crab, Portunus trituberculatus"

_animals, 2023, doi:10.3390/ani13203220_

Round 1

Reviewer 1 Report

The research describes how several biochemical compounds vary between different ovary maturation stages of the Portunus trituberculatus. The sample collection and analysis is well described in the materials and method section. The discussion is well structured and well written. However, I have several main remarks:

-   English revision is necessary. Often sentences are too long or lack the correct verb.

-      Information lacks on the importance and/or relevance of this study.

- It could have been interesting and relevant to also include essential minerals. Why was this not included?

-        All Latin names have to be written in italic

Abstract

Structure is missing in abstract. Explain clearly, why and how the research was performed. Then the results can be discussed and the relevance of the results.

L11  ‘its mature ovary often determine the commercial value and production.’ Explain why (possibly in the introduction)..

L10 ‘most important economic species’ where? Globally? In China?

L14 specify which analyse were performed

L14 specify method and data analysis

Introduction

A clear description of the hypotheses or research goals is missing. It is, for example, not clear if identifying the different maturation stages was part of the research goals? If so, why is this novel?

L39 language check of sentence

Materials and methods

L137 ‘data’ was analysed. Be more specific, which data and with which objectives were those statistical tests performed.

 English revision is necessary. Often sentences are too long or lack the correct verb.

Author Response

Dear reviewer,

We deeply appreciate the time and efforts you have spent in reviewing our manuscript. The comments are valuable and helpful for revising and improving our paper. We have carefully made corrections, and hoping meet with approval. Revised portions are shown as underlines in the revised version of the manuscript. The correction in the paper and the responses to your comments are as follows:

The research describes how several biochemical compounds vary between different ovary maturation stages of the Portunus trituberculatus. The sample collection and analysis is well described in the materials and method section. The discussion is well structured and well written. However, I have several main remarks:

English revision is necessary. Often sentences are too long or lack the correct verb.

Response: Thanks for your advice. We have asked Dr. Ishara Perera, who is an English native speaker and the expert in research field of animal biology and ecology from Hiroshima University, for professional language editing.

Information lacks on the importance and/or relevance of this study.

Response: We have involved the information on the importance and relevance of this study in the introduction part. Please see lines 48-57 and 80-85.

It could have been interesting and relevant to also include essential minerals. Why was this not included?

Response: Thanks for your good suggestion. Indeed, some essential minerals are likely vital to physiological and biochemical functions and to maintain the normal life processes of aquatic animals. But there are many types of minerals in the sea. Since most of them are often stable, their contents are usually less changed with time at the same site. Moreover, previous study suggested that a relatively high proportion of minerals intake in crabs are from the ambient sea water. Since we collected the crabs from the same study site, which indicate these animals may have similar mineral types and contents in the body even among different sampling periods. In addition, the main purpose of this study is to explore the biochemical compositions of crab, including amino acids and fatty acids etc., which have been proved to change greatly during ovarian developmental stages. However, there are few studies reporting the impact of minerals on the ovarian development of crustaceans, and there are also few reports on the relevant research and technical methods. Taken together, it is likely to be difficult to associate the changes of minerals with ovary development and thus we did not include the minerals in present study. Thanks again for your good suggestion. We will also seriously consider to conduct some achievable research to clarify the relationship between essential minerals and ovarian development in the future.

All Latin names have to be written in italic

Response: We are sorry for the careless. We have modified the Latin name format to italic. Please see lines 47, 96, 323, 335, 326, 335 and 341.

Abstract

Structure is missing in abstract. Explain clearly, why and how the research was performed. Then the results can be discussed and the relevance of the results.

Response: Thanks for your advice. We have restructured the abstract, please see lines 22-29.

L11  ‘its mature ovary often determine the commercial value and production.’ Explain why (possibly in the introduction).

Response: Please see lines 44-45 and 48-54.

L10 ‘most important economic species’ where? Globally? In China?

Response: We have inserted “in China” in the abstract. Please see line 22.

L14 specify which analyse were performed. L14 specify method and data analysis

Response: Please see lines 25-28.

Introduction

A clear description of the hypotheses or research goals is missing. It is, for example, not clear if identifying the different maturation stages was part of the research goals? If so, why is this novel?

Response: Sorry for making you confused. Identifying the different maturation stages is not our main research goals. Our purpose is to figure out the specific nutritional needs of ovary maturation of P. trituberculatus. We have rewritten the research goals at the end of the introduction. Please see lines 82-87.

L39 language check of sentence

Response: Ok, please see lines 52-54.

Materials and methods

L137 ‘data’ was analysed. Be more specific, which data and with which objectives were those statistical tests performed.

Response: Thanks for your valuable advice. We have made the data analysis more specific. Please see lines 154-164.

We have tried our best to improve the manuscript. And some changes were also made in the manuscript but will not influence the content and framework of the paper, and we did not list the changes but showed underline in the revised manuscript.

We sincerely appreciate for the your warm work, hoping the correction will meet with approval.

Thanks again for your comments and suggestions.

Best regards,

Wenping Feng

Reviewer 2 Report

This manuscript addresses changes in the nutrient composition of the ovary, hepatopancreas, and muscle tissues during ovarian development in females of the swimming crab Portunus trituberculatus. This species has a wide distribution through China, Japan and Korea, and is an economically valued fishery species in China, although recent commercial harvests have resulted in a decline in wild populations. Aquaculture efforts are being used to replenish wild stocks, but in order to be successful in aquaculture efforts, it is necessary to understand the nutritional and energetic needs of reproduction.

This study focuses on the nutritional requirements during various stages of ovarian development in P. trituberculatus. Inadequate nutrition can lead to suppression of ovarian development. The authors examined the biochemical composition of the ovary, hepatopancreas, and muscle tissue during ovarian maturation with specific analyses of lipid fractionation, fatty acids, amino acids, and carotenoids. Swimming crabs were collected from the East China Sea and each crab was characterized for growth traits, including carapace length (CL), carapace width (CW), body height (BH), and wet body weight (BW). Each crab was then dissected, and ovary and hepatopancreas were weighed, and the gonadal somatic index (GSI, %) and the hepatopancreas index (HIS, %) were calculated. Appropriate statistical analyses were included in the study design, and data presentation is clear throughout the manuscript.

Biochemical analyses were conducted on various stages of ovarian development. Five stages of ovarian maturation were identified by color and size of the ovary. Results indicate that during the five stages of ovarian development, protein content of the hepatopancreas decreases, while protein and lipid levels in the ovary increased during ovarian maturation. There are also significant changes in the amino acid composition of the ovary during maturation. Changes in fatty acid composition and triglyceride, cholesterol, and phospholipid content were also observed during ovarian maturation. The authors present a very comprehensive discussion on the significance of these results on ovarian maturation and the importance of diet in providing the essential requirements for growth and reproduction. Their data also provide the building blocks for developing suitable diets for brood stock development of P. trituberculatus in aquaculture settings.

This is a well-written manuscript and I recommend it for publication.

Well-written, minor edits for clarity

Author Response

Dear reviewer, 

Thanks so much for your comment. 

We sincerely appreciate for your warm work.

Best regards,

Feng